# RACC: Retrieval-Augmented KV Cache Compression in Long-Context Generation

## Abstract

Large Language Models (LLMs) have achieved remarkable progress in long-context generation. As the context length increases, the Key–Value (KV) cache requires the GPU memory with a linear growth rate. KV cache compression is treated as a promising method to reduce the memory usage by permanently discarding a large portion of unimportant KV pairs, but at the expense of inference accuracy. On the other hand, retrieval-based methods employ the CPU memory to store the full KV cache and compute the attention via expensive CPU-GPU I/O, which keeps the accuracy but suffers from huge inference latency. To address these issues, we propose a new inference framework called RACC, which combines both compression based methods and retrieval based methods. To be specific, we employ the KV cache compression method to maintain a high-quality KV cache in the GPU memory, while sotring all the KV pairs evicted by the compression method. In addition, efficient and accurate retrieval conducted on the CPU side finds out important tokens for the one being generated, which is then concatenated with those KV cached in the GPU memory for accurate generation. Extensive experiments demonstrate that RACC achieves near-lossless inference while using only 15% of the original KV cache. Moreover, its combination with prefill-only compression methods improves generation accuracy by 3–10%. Our code is publicly available at `https://anonymous.4open.science/r/CDKEY/`.

## 1 Introduction

Large language models (LLMs) have exhibited powerful capabilities in long-context generation (Zhao et al., 2023), and have achieved superior performance in scenarios such as multi-turn dialogue (Yi et al., 2024), code comprehension (Denny et al., 2024), and document summarization (Zhang et al., 2024). Models such as Gemini 2.5 Pro (Comanici et al., 2025) and Grok-3 (x.ai, 2025) support input context up to 1 million tokens. On the other hand, Claude 3.7 (Anthropic, 2025) and GPT-5 (OpenAI, 2025) have been able to generate outputs up to 128K tokens. Such an increase in the context of LLM inference leads to a significant burden on the GPU memory, because it has to store the KV cache of size linear to the context length. Considering PaLM-540B with a batch size of 512 and a context length of 2048, the KV cache occupies 3 TB of memory in total, which is three times larger than the model parameters(Pope et al., 2023). Subsequently, the KV cache is an unavoidable bottleneck in long-context generation. Hence, it is crucial to optimize the KV cache without additional model training in memory-constrained inference scenarios.

To make full use of the limited GPU memory, a bulk of works have been proposed in the literature to store only a small part of important KV pairs in GPU memory. Those methods were built on the basis that only a small fraction of the KV cache is critical for generating new tokens (Zhang et al., 2023; Liu et al., 2023). Those KV cache reduction methods could be divided into two categories, i.e., (1) compression-based methods and (2) retrieval-based methods. The former methods (Li et al., 2024; Cai et al., 2024) compress the large-size KV cache into a small subset by permanently discarding the KV pairs of unimportant tokens, which are considered as less important for the subsequently generated tokens. To be specific, they estimate the importance of tokens according to their attention scores w.r.t. the tokens being generated and permanently discard the KV pairs of those tokens with lower attention scores to reduce memory usage. However, the discarded KV pairs may be important for the tokens in the subsequent generation, which presents a negative effect on the inference performance. In contrast, the latter methods (Zhang et al., 2025; Liu et al., 2024) just evict those KV pairs

Figure 1: Illustration of the three modes.

to the CPU memory instead of discarding them directly and will fetch those KV pairs of important tokens for the current token generation via vector retrieval methods. Without KV pairs discarded, those methods retain high accuracy, but suffer from lower efficiency caused by the GPU–CPU I/O bottleneck. As a result, existing methods present a poor balance between accuracy and efficiency.

To address these issues, we propose a new framework called **RACC (Retrieval-Augmented KV Cache Compression)**, which combines both KV cache compression and KV cache retrieval. The three modes are illustrated in Figure 1.

Like KV cache compression, RACC maintains a compressed KV cache in GPU memory, which consists of a subset of important KV pairs from the full KV cache. Unlike KV cache compression that overlooks the KV pairs evicted previously, RACC employs a vector retrieval module maintained on the CPU side, which manages all evicted KV pairs and retrieves potentially important KV pairs to the currently generated token. Still, our framework has to deal with two challenges, i.e., (1) the selection of tokens as the query in vector retrieval and (2) the vector retrieval method on the CPU side. Existing temporary eviction methods use the current token to be generated as the query, which degrades the whole inference latency. This is because the inference has to wait for the results of the vector retrieval, which has to communicate with the lowly efficient CPU via GPU-CPU I/O. In this work, we propose to employ the recently generated tokens instead of the currently generated one as the query for vector retrieval, on the basis of the observation that the adjacent tokens in the sequence have similar attention tokens to the currently generated token. In this way, we asynchronously conduct the vector retrieval with the previously generated tokens as the queries and then send the retrieval results to the GPU as a part of KV cache for the current token. On the other hand, the vector retrieval must be efficient and accurate so that the retrieval results will be used for the generation of the tokens near the queries.

We conducted extensive experiments to evaluate the effectiveness of our approach. We selected two benchmark tests, Longgenbench (Wu et al., 2025)and Longproc (Ye et al., 2025), to assess the performance of our method, comparing it with the current state-of-the-art KV compression techniques. The results show that **RACC** achieves the lowest inference latency while maintaining nearly lossless performance with only 15% of the original KV cache. We also designed experiments to combine **RACC**'s retrieval module with other KV compression methods, using Longbench(Bai et al., 2023) as a benchmark to evaluate their generation accuracy. Compared to their original performance, we achieved an improvement of 3%-10% in accuracy scores.

Our contributions are summarized as follows:

1. We propose a new framework of KV cache compression that combines both permanent eviction and temporary eviction.

2. We carefully design the core modules of our framework, i.e., query selection and vector retrieval.

3. We conduct experiments on widely-used benchmarks to demonstrate the effectiveness and advantages of our method.

## 2 BACKGROUND AND RELATED WORKS

### 2.1 KV CACHE IN LLM INFERENCE

LLM inference can be divided into two phases: the *prefilling* phase and the *decoding* phase.

In the Prefilling phase, the user's input prompt is embedded into word vectors, followed by the addition of positional encoding. These representations are then linearly projected into the query, key, and value matrices, denoted as

$$Q^{\text{prompt}} \in \mathbb{R}^{B \times S \times H \times D}, \quad K^{\text{prompt}} \in \mathbb{R}^{B \times S \times H \times D}, \quad V^{\text{prompt}} \in \mathbb{R}^{B \times S \times H \times D},$$

where $B$ is the batch size, $S$ is the sequence length, $H$ is the number of attention heads, and $D$ is the dimensionality of each head.

After the matrix operations in Eq. 1 (Vaswani et al., 2017), the output of a single-layer self-attention module is produced and subsequently fed into the next transformer layer for forward propagation. After all layers of propagation, all $K$ and $V$ matrices have been computed. These matrices are stored as the *KV cache*, which is then utilized in the decoding phase to generate the first predicted token.

$$\text{Attention}(Q, K, V) = \text{Softmax}\left(\frac{QK^{\top}}{\sqrt{D}}\right) V \tag{1}$$

In the decoding phase, each forward propagation step generates one new token in an auto-regressive manner until an end-of-sequence symbol is produced. The previously generated token is projected into

$$Q^{\text{new}} \in \mathbb{R}^{B \times 1 \times H \times D}, \quad K^{\text{new}} \in \mathbb{R}^{B \times 1 \times H \times D}, \quad V^{\text{new}} \in \mathbb{R}^{B \times 1 \times H \times D}.$$

At this point, the newly generated $K$ and $V$ are concatenated with the cached KV matrices from the prefilling phase, forming the updated KV cache as shown in Eq. 2:

$$K = \text{Concat}(K^{\text{prompt}}, K^{\text{new}}, \dim = 1), \quad V = \text{Concat}(V^{\text{prompt}}, V^{\text{new}}, \dim = 1), \tag{2}$$

which results in

$$K \in \mathbb{R}^{B \times (S+1) \times H \times D}, \quad V \in \mathbb{R}^{B \times (S+1) \times H \times D}.$$

The updated $K$ and $V$ are then combined with the query $Q$ to perform the attention computation as in Eq. 1, followed by forward propagation to predict the next token.

During generation, the memory footprint of the KV cache grows linearly with the length of the generated sequence. If there is either an excessively long input prompt or output, this results in substantial memory consumption or even the OOM error. The issue becomes worse in batch inference scenarios, where the memory overhead is amplified across multiple sequences. Consequently, effective compression strategies for the KV cache are urgently needed.

## 2.2 RELATED WORKS

To extend the ability of LLM in long-context generation with limited GPU memory, a bulk of methods have been proposed, which could be divided into two categories, i.e., (1) vector retrieval based methods and (2) KV cache compression based methods. Both of them are based on the fact that only a small subset of essential tokens exerts a predominant influence on the generation fidelity (Zhang et al., 2023; Liu et al., 2023).

Vector retrieval methods (Liu et al., 2024; Zhang et al., 2025) usually employ the CPU memory to help store the whole KV cache and identify the essential tokens via vector retrieval. RetrievalAttention (Liu et al., 2024) initially offloads the entire KV cache to the CPU and performs online retrieval of the essential tokens for the current one. It enables lossless inference, but suffers from unacceptable generation latency due to the online retrieval via the brute-force method. To accelerate the CPU-side retrieval, PQCache (Zhang et al., 2025) employs the SOTA vector retrieval method, i.e., product quantization (PQ) (Jegou et al., 2010), which is efficient but only returns approximate retrieval results. Hence, the approximate retrieval results affect the subsequent generation accuracy. Besides, PQCache still suffers from long generation latency due to the expensive data transfer between CPU and GPU in an online manner.

KV cache compression methods do not store the full KV cache and thus do not need the cooperation of the CPU. They only retain the identified essential tokens, which are then used for subsequent generations, while permanently discarding others. As a representative, SnapKV (Li et al., 2024) observes that attention allocation patterns remain stable in the generation phase, and thus identifies the critical prompt tokens using a sliding window at the tail of the prompt. PyramidKV (Cai et al.,

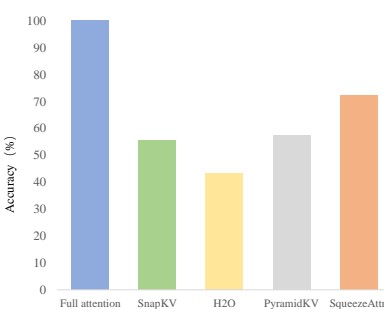 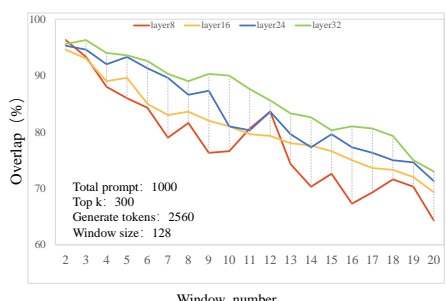

Figure 2: Accuracy evaluation on LongGen-Bench, comparing uncompressed inference with various KV compression methods.

Figure 3: The overlap rate between the top-$k$ tokens selected from window 1 and those selected from the subsequent 19 windows.

2024) argues that different model layers exhibit distinct attention-allocation patterns and thus assigns layer-wise memory budgets to each KV cache to enhance the KV cache compression. Cake(Qin et al., 2025) further analyzes the attention preferences across different layers to adaptively allocate the memory budget of the KV cache. SqueezeAttention(Wang et al., 2024) employs a sliding-window mechanism spanning both the Prefilling and Decoding phases, compressing the KV cache by discarding entries outside the window. Similarly, ScopeWu et al. (2024) adopts a sliding-window strategy but introduces a more flexible eviction policy for KV entries. Even though KV cache compression methods achieve a good compression ratio and almost keep the inference speed, they still suffer from significant accuracy loss due to their permanent discarding policy.

## 3 OBSERVATIONS

In this section, we present our observations on attention allocation in generation as listed in the following. Our experiments are conducted on two long-generation benchmarks, i.e., LongProc(Ye et al., 2025) and LongGenBench(Wu et al., 2025), with Meta-Llama-3.1-8B-Instruct as the LLM.

1. **KV Cache compression methods suffer from significant accuracy loss in long-generation scenarios.** We evaluate several representative KV cache compression methods on the LongGenBench dataset, ensuring that they operate under the same compression ratio, and examine their accuracy scores. As shown in Figure 2, where we use four representative KV cache compression methods (SnapKV (Li et al., 2024), H2O (Zhang et al., 2023), PyramidKV (Cai et al., 2024) and SequeezeAttn (Wang et al., 2024)), we can see that all compression methods obviously decrease their accuracy compared with the full attention in the long-generation senarios. This observation indicates a part of important tokens were permanently discarded by the compression methods and has no chance to recall them back.

2. **Adjacent tokens generated in the decoding phase present highly similar attention allocation patterns.** Our experiments are based on the "Travel Planning" dataset from Long-Proc in long-input ($\geq$ 4K tokens) long-output (2K-8K tokens). We divide the generated tokens at each layer into windows of 100 tokens and select the first 20 windows in the decoding phase. For each selected window, we observe its attention allocation w.r.t. the prompt tokens. We compute the overlap ratio between the tokens identified with the highest attention scores in the first window and those in subsequent windows. As in Figure 3, we observe that adjacent windows share similar attention allocation patterns, whereas this similarity decreases as the window distance increases.

## 4 METHODOLOGY

In this section, we present our method RACC that combines both KV cache compression and vector retrieval on the basis of the two observations in the last section. The intuition of our method is to find important KV pairs for the current token from the evicted KV pairs by the compression methods via

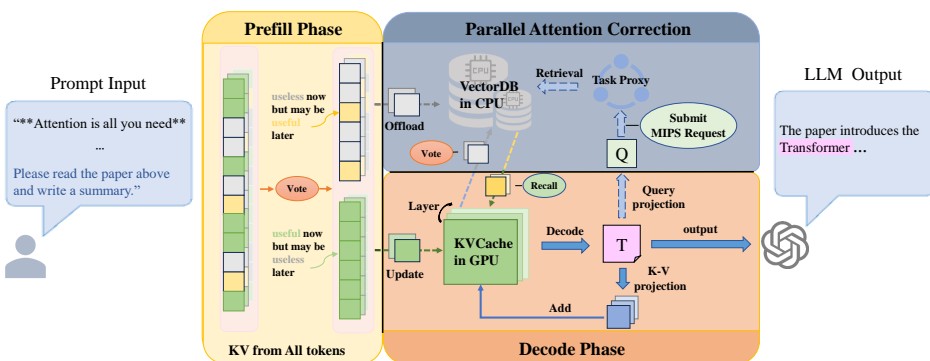

Figure 4: Illustrating the RACC framework.

the asynchronous CPU-side vector retrieval. In this way, we enhance the accuracy of compression methods while not compromising both the efficiency and the compression ratio compared with the compression-only methods.

## 4.1 OVERVIEW

Existing compression methods produce accurate generation for a short output after compression, but degrade gradually for a long output due to the accumulated errors(Sun et al., 2025; Choi et al., 2025). This is an inherent limitation of all KV compression methods, where the discarded KV may still contain crucial information for subsequent generations. To address this issue, it is necessary to manage evicted KV by those compression methods and inquire them again in subsequent generations.

As illustrated in Fig. 4, RACC consists of two key components, i.e., (1) a retrieval module that runs on the CPU side in parallel with the GPU and (2) a KV cache compression module that maintains a subset of essential KV pairs in the GPU memory. In the retrieval module, we manage the KV pairs evicted by the compression module and build a vector index for efficient and accurate retrieval. This module can identify important tokens beyond the compressed KV cache in the GPU memory and then sends the KV pairs of those tokens found back to the GPU memory, which will be used in the following token generations. In the compression module, we employ existing KV cache compression methods to obtain the compressed KV pairs online. Notably, our framework is compatible to all KV cache compression methods and vector retrieval methods. Moreover, we design a fine-grained scheduling strategy to tightly coordinate the two modules, ensuring that inference proceeds without incurring additional latency.

## 4.2 THE RETRIEVAL MODULE

Our overall objective is to *recycle* a small query-dependent subset of evicted KV to correct the attention computation, which better approximates the full-context attention. To ensure GPU inference efficiency, this correction computation should not be performed on the GPU and also not block the GPU generation.

**CPU–Side KV Retrieval.** In general, RACC decouples *computation* and *retrieval* by retaining discarded KV pairs in CPU memory and performing retrieval asynchronously on the CPU side, which does not interrupt the GPU inference.

- After each KV compression, evicted KV pairs are offloaded to CPU memory and then managed by a vector index for efficient and accurate vector retrieval.

- For each query $q_t$, the GPU sends a vector retrieval request to the CPU, which finds relevant KV pairs from evicted ones asynchronously.

- Once the GPU receives the retrieval result from the CPU, it directly concatenates them with the current KV cache, which will be used in the subsequent token generations.

**CPU-side retrieval is in fact maximum inner product search (MIPS).** Let us consider the attention distribution at each decoding step $t$. Formally, let $KV_R$ ($K_R$) denote the current KV (keys) cache retained on the GPU and $KV_E$ ($K_E$) the evicted KV (keys) by the compression methods. For the current query $q_t$, the ideal attention distribution over the complete context $K_R \cup K_E$ is

$$p_i^\star = \frac{\exp\left(\langle q_t, k_i\rangle / \sqrt{D}\right)}{\sum_{j \in K_R \cup K_E} \exp\left(\langle q_t, k_j\rangle / \sqrt{D}\right)}, \quad i \in K_R \cup K_E \tag{3}$$

Here, $q_t$ represents the query vector at decoding step $t$, and $k_i$ refers to the $i$-th key vector in the context set, $D$ denotes the dimensionality of the vectors. Since the attention weight in Eq. 3 grows with the inner product $\langle q_t, k_i\rangle$, selecting the most relevant evicted pairs reduces to a Maximum Inner Product Search (MIPS) problem with $q_t$ as query and returns a subset $K_E' \subset K_E$ such that $\forall k_1 \in K_E', k_2 \in K_E \setminus K_E', \langle q_t, k_1\rangle \geq \langle q_t, k_2\rangle$ holds.

Efficient and accurate MIPS remains a long-standing challenge ((Indyk & Motwani, 1998; Chen, 2018)). Therefore, we resort to the approximate MIPS, which accelerates the retrieval at the expense of accuracy. To be specific, approximate MIPS methods (Guo et al. (2016); Bruch et al. (2023); Chen et al. (2025)) enhance the search efficiency at the expense of the search accuracy. Moreover, through vector space transformations (Bachrach et al. (2014); Zhou et al. (2019)), the MIPS problem can be reformulated as a standard nearest neighbor search (NNS) problem. We will provide the proof of this process in the appendix D. Such a transformation enables RACC to leverage standard NNS methods (Jegou et al. (2010); Malkov & Yashunin (2018)), which have already been adopted by mature vector databases ((Wang et al., 2021; Douze et al., 2024)).

**The query $q_t$ in MIPS is not the token being generated in the GPU.** Since the CPU-side retrieval cannot block the GPU inference, the MIPS query must not be the token being generated. Owing to the second observation in Section 3, adjacent tokens have similar attention allocation. Hence, we use the token generated before the one being generated as the query of MIPS retrieval. In this way, our method RACC captures the attention allocation better than the compression-only method, and is more efficient than the retrieval-only methods that issue a synchronous retrieval.

### 4.3 THE COMPRESSION MODULE

**Different scenarios should take different compression strategies.** We design an independent compression module for RACC that can adapt to various compression scenarios. Specifically, we categorize long-context tasks into three types: *long-input short-output (LISO)*, *short-input long-output (SILO)*, and *long-input long-output (LILO)*. Most existing compression methods are typically tailored to only one of them. In contrast, our compression module applies distinct strategies to each scenario and constructs different indexes accordingly. SnapKV (Li et al., 2024) was the first to introduce the use of an **observation window** to perform **voting**, selecting important past KV pairs for KV cache compression. We follow this idea in our framework, and adopt the following observation window selection strategies.

1. **LISO:** In the decoding phase, the attention pattern remains stable. We select the last segment of the prompt as the observation window to perform voting over all past KV pairs, while caching all the KV pairs generated in the decoding phase.

2. **SILO:** Since the prompt is short and the attention pattern changes frequently in decoding, we select the most recent window of generated tokens as the observation window. Voting is then applied to compress all previously generated tokens, while retaining all KV pairs from the prefilling phase.

3. **LILO:** We apply the above strategies to compress KV pairs from both the prefilling and decoding phases.

The illustrations of the three strategies is shown in Fig. 5. The mathematical formulations for the compression window and voting mechanism are provided in the Appendix B.

After obtaining the voting scores, we selectively retain the most important past tokens. Existing methods (Li et al., 2024; Cai et al., 2024; Wang et al., 2024) limit the KV cache budget according to the compression ratio and select the KV entries with the highest scores. Once the cached KV pairs

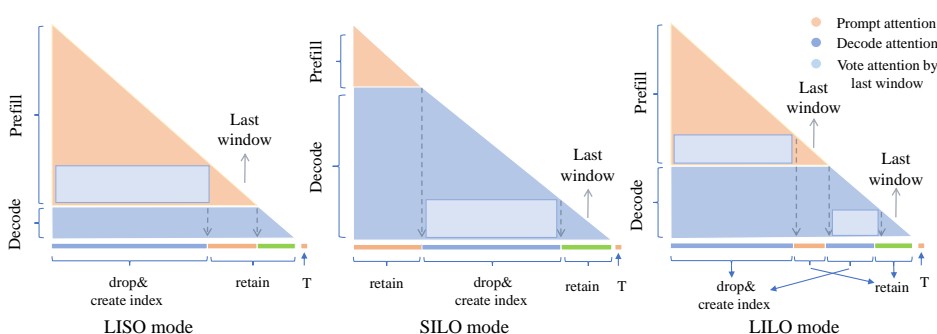

Figure 5: Illustration of the vote distribution under three different compression modes.

exceed the KV cache budget, frequent voting computations are required to guarantee the constraint of the KV cache budget.

To address this issue, we adopt a grouping compression strategy, which is conducted for every $L_n$ newly generated tokens instead of every one. Specifically, once $L_n$ new tokens are generated, we perform a compression operation over all KV entries. The compression is controlled by a tunable hyperparameter $\alpha$. Before compression, the length of the KV cache is denoted as $L_{\text{history}}$. After each compression, the resulting KV length becomes $L_{\text{compression}}$.

$$L_{\text{compression}} = (L_{\text{history}} + L_n) \times \alpha.$$

This grouping strategy allows the KV budget to be dynamically adjusted and significantly reduces the number of expensive compression operations. Hence, it enhances the overall efficiency of the GPU inference. Even though fewer KV cache updates in the generation, our method does not compromise the quality of the KV cache, because close tokens in the generation sequence have similar attention patterns as discussed in the second observation in Section 3. The effectiveness of such a grouping compression strategy is verified in the experiments. For more details, the pseudocode of the compression strategy is provided in the Appendix C.

**Different index construction strategies are applied for different compression scenarios.** In the LISO scenario, the aforementioned compression strategy is used to offload unimportant KV entries to the CPU and construct a static index. As to SILO and LILO scenarios, when evicted KV entries are offloaded to the CPU for the first time, a dynamic index is constructed. Subsequently, as more KV entries are offloaded, the index is incrementally updated, i.e., new KV entries are dynamically inserted into the existing index.

## 5 EXPERIMENTS

### 5.1 EXPERIMENTAL SETUP

**Datasets:** We evaluate our method on two open-source datasets: **LongGenBench**(Wu et al., 2025) and **LongProc**(Ye et al., 2025). Both datasets are designed to assess the model's accuracy in long-form generation tasks. For **LongGenBench**, we select a generation length of 8K tokens for the short dataset and 16K tokens for the long dataset. For **LongProc**, the generation length is dynamically determined based on the specific sub-dataset. Additionally, we employ retrieval techniques in combination with **SnapKV** and **PyramidKV** in an orthogonal manner to further improve accuracy. We utilize **LongBench**(Bai et al., 2023) to evaluate the effectiveness of this integration. Detailed descriptions of dataset selection and configuration can be found in the Appendix A.

**Baseline Methods:** To evaluate the effectiveness of RACC, we compare it against full-cache generation and several other compression-based methods.

1. **Scope**(Wu et al., 2024) performs KV compression in the decoding stage in a sliding manner.

2. **CakeKV**(Qin et al., 2025) focuses on the allocation of KV cache size among layers.

3. **SqueezeAttn**(Wang et al., 2024) adjusts the sliding window budget across layers.

Table 1: Latency comparison of RACC and baseline methods.

| Method | TTFT (ms) | OGL with Different Generated Tokens | | |
|---|---|---|---|---|
| | | 512 (s) | 1024 (s) | 2048 (s) |
| Full Cache | 254 | 15.40 | 30.71 | 63.88 |
| SqueezeAttn | 275 | 20.84 | 40.98 | 81.75 |
| Scope | 413 | 19.27 | 38.24 | 83.88 |
| Cakekv | 268 | 16.23 | 33.85 | 70.72 |
| RACC | 259 | 15.96 | 32.40 | 65.66 |

Table 2: Comparison of Generation Accuracy in LongGenBench

| Model | Method | Short Experiment (token=8k) | | | Long Experiment (token=16k) | | |
|---|---|---|---|---|---|---|---|
| | | Completion Rate | Single Accuracy | Range Accuracy | Completion Rate | Single Accuracy | Range Accuracy |
| Llama | Full Cache | 66.0 | 34.7 | 58.2 | 37.5 | 60.8 | 53.4 |
| | SqueezeAttn | 61.8 | 29.7 | 52.3 | 34.3 | 56.0 | 48.4 |
| | Scope | 62.0 | 27.3 | 52.1 | 34.8 | 57.1 | 43.7 |
| | CakeKV | 59.5 | 25.8 | 39.8 | 32.5 | 54.6 | 39.8 |
| | RACC | 65.8 | 31.6 | 54.5 | 36.0 | 58.3 | 50.5 |
| Mistral | Full Cache | 67.5 | 39.1 | 62.4 | 36.5 | 56.1 | 52.7 |
| | SqueezeAttn | 64.3 | 36.0 | 48.4 | 34.3 | 46.2 | 44.7 |
| | Scope | 64.8 | 36.5 | 55.7 | 32.8 | 46.5 | 45.7 |
| | CakeKV | 62.5 | 24.6 | 32.8 | 32.5 | 44.6 | 42.8 |
| | RACC | 66.0 | 38.0 | 58.5 | 35.0 | 48.4 | 48.5 |

Notably, our method **RACC** is able to seamlessly integrate with any KV cache compression methods in addition to **SnapKV** and **PyramidKV**.

**Implementation Details:** We conduct our experiments on a NVIDIA A6000 GPU and an Intel(R) Xeon(R) Gold 6240 CPU @ 2.60GHz. We use **LLaMA-3.1-8B** and **Mistral-7B-v0.2** as the LLM models, both of which demonstrate strong performance on the aforementioned datasets and support long generation. Our method sets the compression ratio of KV cache compression as a hyperparameter, dynamically compressing tokens in generation. In this work, we define the compression ratio as the ratio between the number of tokens stored in GPU at the last generation step and the total number of generated tokens. More details could be found in the Appendix A.

## 5.2 GENERATION LATENCY AND FIRST-TOKEN LATENCY

Two metrics, i.e., overall generation latency (OGL) and time to first token (TTFT), are used to evaluate the generation latency. We show our experimental results in Table 1, where Full Cache refers to inference without any KV compression. First, KV cache compression methods introduce additional cost, and thus present significantly higher generation latency than full cache in SILO scenarios. Among the methods with KV cache compression, our method RACC achieves the smallest generation latency, and is even comparable to full cache. This experiment demonstrates the advantages of our method in generation latency.

## 5.3 GENERATION ACCURACY EVALUATION

We further compare our method with baselines in generation accuracy. To make a fair comparison, we allocate the same memory budget of KV Cache for all compression methods by setting their compression ratio to 15% during inference. The experimental results in LongGenBench and Long-Proc are shown in Table 2 and Table 3, respectively. Here, "completion rate" denotes the proportion of cases where the model produces outputs that follow the required prompt format, while "accuracy" measures the proportion of correct responses among them. The results demonstrate that, with the same KV cache budget, our method, RACC, presents significantly higher generation accuracy than

Table 3: Comparison between RACC and the baseline across different datasets in LongProc

| Model | Method | Path traversal (8K)) | | Tom tracking (8K) | | Travel planning (8k) | |
|---|---|---|---|---|---|---|---|
| | | Completion Rate | Task Accuracy | Completion Rate | Task Accuracy | Completion Rate | Task Accuracy |
| Llama | Full Cache | 66.7 | 1.99 | 100.0 | 18.42 | 53.3 | 5.31 |
| | SqueezeAttn | 63.3 | 1.70 | 100.0 | 17.40 | 43.3 | 5.21 |
| | Scope | 60.0 | 1.84 | 100.0 | 16.34 | 46.7 | 5.15 |
| | CakeKV | 63.3 | 1.65 | 80.0 | 15.17 | 39.6 | 3.48 |
| | RACC | 66.3 | 1.73 | 100.0 | 17.40 | 52.7 | 5.25 |
| Mistral | Full Cache | 74.5 | 2.01 | 100.0 | 15.67 | 52.7 | 5.38 |
| | SqueezeAttn | 67.4 | 1.78 | 80.0 | 13.27 | 43.5 | 4.73 |
| | Scope | 65.8 | 1.83 | 100.0 | 14.20 | 47.7 | 4.67 |
| | CakeKV | 58.5 | 1.71 | 80.0 | 13.67 | 42.6 | 3.98 |
| | RACC | 68.0 | 1.89 | 100.0 | 14.60 | 47.5 | 5.13 |

Table 4: Performance (Score) of different methods across Longbench for Llama and Mistral.

| Method | Llama | | | Mistral | | |
|---|---|---|---|---|---|---|
| | gov_report | muti_news | qmsum | gov_report | muti_news | qmsum |
| SnapKV | 26.84 | 22.09 | 22.48 | 30.20 | 24.36 | 24.71 |
| SnapKV + RACC | 27.95 | 24.2 | 23.35 | 31.14 | 25.47 | 25.53 |
| PyramidKV | 27.84 | 21.75 | 23.63 | 27.59 | 22.31 | 23.96 |
| PyramidKV + RACC | 28.70 | 22.38 | 24.46 | 28.40 | 22.95 | 24.67 |

the baseline compression methods. Such an improvement in accuracy is achieved by the retrieval of essential tokens from the evicted ones through the compression method. In particular, our CPU-side vector retrieval is efficient and accurate, and executed in an asynchronous manner that does not block the GPU inference. Additional experimental results under different parameter settings can be found in Appendix E.

## 5.4 TESTING THE GENERALITY OF RACC

As aforementioned, our method RACC could be seamlessly integrated with the KV cache compression method. In the previous experiments, we focus on the SILO benchmarks, where the decoding-stage KV cache compression is crucial. In this experiment, we integrate our method with prefilling-only compression methods, i.e., SnapKV and PyramidKV on LISO benchmark LongBench. The experimental results are reported in Table 4. We can see that SnapKV + RACC obviously outperforms SnapKV in various tasks, achieving 3%–9.6% improvement in accuracy. Besides, similar phenomena could be found between PyramidKV and PyramidKV + RACC. This experiment demonstrates the generality of our RACC framework that is able to seamlessly integrate with any KV cache compression method.

## 6 CONCLUSION

In this work, we aim to enhance the performance of LLM inference with a limited GPU memory budget. We propose a combined method that integrates KV cache compression methods and CPU-side vector retrieval methods. Moreover, we employ the closes tokens of the one being generated in the output sequence as the query in the vector retrieval and execute the retrieval in an asynchronous manner, which maintains the GPU inference speed and also returns high-quality KV pairs for the current token. Our experiments demonstrate the superiority of our method in both efficiency and accuracy over existing KV cache compreison methods.

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

# A EXPERIMENTAL DETAILS

## A.1 METHOD FOR CALCULATING COMPRESSION RATIOS

Different KV compression works adopt varying strategies for KV compression. Some works do not directly treat "compression ratio" as a hyperparameter, but instead set a memory budget for KV at each layer or dynamically adjust the compression ratio. Therefore, we propose the following calculation rule: the compression ratio $P$ is defined as the ratio of the number of KV pairs stored in the GPU at the end of the model's generation to the total number of KV pairs in the model's context. Since the number of KVs is equivalent to the number of tokens, we will use the token count to refer to the tokens. For **CakeKV**(Qin et al., 2025), **Scope**(Wu et al., 2024), and **SqueezeAttn**(Wang et al., 2024), KV cache compression is performed by fixing the number of tokens allocated (budgeted) in each layer's KV cache. Therefore, the compression ratio $P$ is calculated as the ratio of the budgeted tokens to the generated tokens.

$$P = \frac{\text{\#Budgeted Tokens}}{\text{\#Generated Tokens}}$$

For **RACC**, compression is performed on the entire KV cache every time $L_n$ new KV pairs are generated. The compression rate for each compression operation is defined as the hyperparameter $\alpha$. Additionally, $\beta$ is defined as the proportion of KV pairs retrieved from the CPU relative to the total KV in the context.

The number of compression operations $n$ is determined by dividing the total number of generated tokens by $L_n$, and rounding down to the nearest integer:

$$n = \left\lfloor \frac{\text{\#Generated tokens}}{L_n} \right\rfloor$$

Therefore, the compression rate $P_R$ is given by the sum of a geometric series, plus the retrieval cost $\beta$:

$$P_R = \frac{\alpha^n - 1}{n(\alpha - 1)} + \beta \tag{4}$$

Here, $\alpha$ is the hyperparameter defining the compression rate for each operation, $n$ is the number of compression operations, and $\beta$ represents the retrieval cost of KV pairs from the CPU.

## A.2 DETAILS OF THE EXPERIMENTAL SETUP

For all the control groups in Experiments 5.2 and 5.3, the full prompt sequence KV is preserved with a compression rate set to 15%. In Eq. 4, we define three parameters related to the compression rate $P_R$, namely $n$, $\alpha$, and $\beta$, where $n$ is determined by the window size $L_n$. We treat $\alpha$, $\beta$, and $L_n$ as hyperparameters. In the experiments presented in the main text, we set $\alpha = 60\%$, $\beta = 1\%$, and $L_n = 400$. A sensitivity analysis of these three parameters is provided in Appendix E.3.

## A.3 DATASET SELECTION

For the accuracy evaluation of long-text generation, we select two benchmarks, LongGenbench and Longproc, which are designed to assess the model's performance in long-text generation. For the experiments in Section 5.4, we choose three datasets from LongBench: gov_report, multi_news, and qmsum. The prompts in LongBench are generally long, making them suitable for evaluating the model's ability to comprehend and utilize context. We did not choose all available datasets for testing, because for other datasets, the output length of the generated text is within the range of 50–200 tokens, where the KV cache selection strategy incurs minimal errors. Therefore, no retrieval techniques are required for error correction. In contrast, the datasets gov_report, multi_news, and qmsum have output lengths of 512 tokens, which provide a sufficiently long context window for retrieval techniques to correct errors. This was also verified in our experiments.

# B DETAILED EXPLANATION OF THE VOTING MECHANISM

As shown in Figure 6, in order to describe the method, we introduce the following terminology:

- **Prompt length** ($L1$): the length of the prompt token sequence.
- **Previously generated tokens** ($L2$): all tokens generated prior to the most recent window, which account for the majority of the sequence.
- **Observation window tokens** ($L3$): all tokens within the most recent window. The total number of tokens before generating the new token is defined as

$$L = L1 \oplus L2 \oplus L3.$$

where $\oplus$ denotes the concatenation operation.

- **New token** ($T$): the token that the model is currently generating.
- **Vote attention**: attention scores are computed between the queries from $L3$ and the keys from $L2$. Formally, given the query $Q_{L3} \in \mathbb{R}^{B \times L3 \times H \times D}$ and the cached keys $K_{L2} \in \mathbb{R}^{B \times L2 \times H \times D}$, the score matrix ($S$) is defined as

$$S_{L3 \to L2} = \frac{Q_{L3} K_{L2}^{\top}}{\sqrt{D}}, \tag{5}$$

where

$$S_{L3 \to L2} \in \mathbb{R}^{B \times H \times L3 \times L2}.$$

By summing over the $L3$ dimension, we obtain the final vote score for each token in $L2$:

$$S_{L2} = \sum_{i=1}^{L3} S_{L3 \to L2}[:,:,i,:], \tag{6}$$

with

$$S_{L2} \in \mathbb{R}^{B \times H \times L2}.$$

Here, $B$ denotes the batch size and $H$ denotes the number of attention heads.

- **Top-k Compression**: Based on the vote scores $s_{L2}$, we select the top-k tokens in $L2$ to compress. This is done by sorting the attention scores and choosing the top-k highest-scoring tokens for each attention head:

$$S_{\text{top-k}} = \text{TopK}(S_{L2}, k),$$

where $S_{\text{top-k}}$ contains the indices of the top-k tokens.

The corresponding tokens in $L2$ are selected based on these indices and compressed.

- **Reconstruction**: After compression, the compressed $L2$ is concatenated back with the original $L1$ and $L3$:

$$L_{\text{final}} = L1 \oplus \text{Compressed}(L2) \oplus L3,$$

where $\oplus$ denotes the concatenation operation.

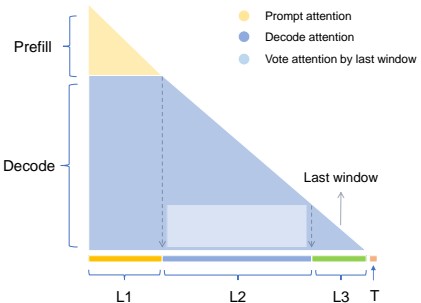

Figure 6: Description of the vote mechanism.

## C    DETAILED PSEUDOCODE FOR THE COMPRESSION STRATEGY

The algorithm 1 presents the pseudocode for our Compression Strategy. Lines 3 to 6 illustrate the *vote attention* from $L3$ to $L2$, which selects the critical token sequences from $L2$. Lines 12 to 26 show the update of the KV cache in GPU memory. Lines 19 to 23 describe an optional retrieval strategy that loads previously discarded tokens—whose keys and values are critical for the current generation—back into the GPU and concatenates them with the existing KV cache.

---

**Algorithm 1** KV Compression with Optional Offline KV Linking

---

1:  **Input:** Key, Value, Query_window, compress_ratio, retrieval_kv (optional), Index (optional)
2:  **Output:** Updated Key and Value
3:  $attn\_weight \leftarrow$ vote_attn($Query\_window, Key$)
4:  $vote \leftarrow attn\_weight$.sum(column-wise)
5:  $K \leftarrow L2\_size \times compress\_ratio$
6:  $topk\_index \leftarrow vote$.topk($K$)
7:  $full\_index \leftarrow \{0, 1, \ldots, L1\_size + L2\_size + L3\_size - 1\}$
8:  $drop\_index \leftarrow full\_index \setminus (L1 \cup L3 \cup topk\_index)$
9:  *// Compute indices of tokens to be dropped*
10:  $compress\_key \leftarrow Key_{L2}$.gather(index $= topk\_index$)
11:  $compress\_value \leftarrow Value_{L2}$.gather(index $= topk\_index$)
12:  $Key \leftarrow$ concat($Key_{L1}, compress\_key, Key_{L3}$)
13:  $Value \leftarrow$ concat($Value_{L1}, compress\_value, Value_{L3}$)
14:  **if** retrieval_kv is not None **then**
15:      *// Link offline retrieval KV*
16:      $Key \leftarrow$ concat($Key, retrieval\_kv.key\_states$)
17:      $Value \leftarrow$ concat($Value, retrieval\_kv.value\_states$)
18:  **end if**
19:  **if** Index is not None **then**
20:      *// Offload dropped KV to CPU and update index*
21:      $drop\_key \leftarrow Key_{L2}$.gather(index $= drop\_index$).to(cpu)
22:      $drop\_value \leftarrow Value_{L2}$.gather(index $= drop\_index$).to(cpu)
23:      $Index$.add($drop\_key, drop\_value$)
24:  **end if**
25:  *// Update KV cache*
26:  $Cache$.update($Key, Value, layer\_id$)
27:  **return** $Key, Value$

---

## D    DETAILED EXPLANATION OF MIPS

### D.1    PROOF OF THE EQUIVALENCE BETWEEN MIPS AND NNS

We now present a formal proof sketch showing that the Maximum Inner Product Search (MIPS) problem can be reduced to a Nearest Neighbor Search (NNS) problem in Euclidean space.

**Problem Setup.**    Given a query $q \in \mathbb{R}^D$ and database $\mathcal{X} \subset \mathbb{R}^D$, MIPS seeks

$$x^* = \arg\max_{x \in \mathcal{X}} \langle q, x \rangle,$$

while NNS under $\ell_2$ norm is

$$x' = \arg\min_{x \in \mathcal{X}} \|q - x\|_2.$$

**Transformation.**    Define

$$\phi(x) = \text{Concat}\left(x, \sqrt{M - \|x\|_2^2}\right), \quad \upsilon(q) = \text{Concat}(q, 0),$$

with $M \geq \max_{x \in \mathcal{X}} \|x\|_2^2$.

**Equivalence.** We have

$$\|v(q) - \phi(x)\|_2^2 = \|q\|_2^2 + M - 2\langle q, x\rangle.$$

Since $\|q\|_2^2$ and $M$ are constants, minimizing the $\ell_2$ distance is equivalent to maximizing $\langle q, x\rangle$.

## D.2 IMPLEMENTATION

The main advantage of transforming MIPS into NNS is that it enables the use of highly optimized nearest neighbor indices, thereby providing a convenient and high-performance solution for inner product search. We implement this nearest neighbor search on top of Faiss(Douze et al., 2024), a widely-used open-source vector library, and adapt it for three types of indexes: Flat Index, IVFPQ(Jegou et al., 2010), and HNSW(Malkov & Yashunin, 2018).

**Flat Index**: The Flat index stores all vectors in a simple list, where the vector search is performed by computing the distance between the query vector and every stored vector to find the most similar one. Unlike other more complex indexing methods, the Flat index does not involve any division or optimization of the vector space. The retrieval process is a linear scan, where each stored vector is compared one by one, returning the closest result.

**IVFPQ (Inverted File with Product Quantization)**: IVFPQ divides the vector space into multiple clusters and stores vectors in their corresponding clusters. During search, it first locates the top-$n$ closest clusters to the query vector, and then searches within these clusters. To further reduce memory usage and accelerate distance computation, the vectors in each cluster are compressed using product quantization (PQ), where each vector is partitioned into sub-vectors and quantized separately. This combination significantly reduces the number of vectors to compare while also lowering storage cost, leading to efficient large-scale retrieval.

**HNSW (Hierarchical Navigable Small World)**: HNSW constructs multi-level graphs to map the vector space, with edges connecting vectors to represent their similarity. The search process in HNSW traverses the graph, maintaining a candidate list to store the top-$k$ nearest vectors. This index is particularly effective for high-dimensional vector spaces, as it balances search accuracy with efficiency through hierarchical graph traversal.

## D.3 EVALUATION OF APPROACHES

We design experiments to evaluate the memory usage, latency, and query accuracy of the three retrieval methods mentioned above.

### EXPERIMENTAL SETUP

We conducted experiments on an Intel(R) Xeon(R) Gold 6240 CPU @ 2.60GHz, using the GloVe (Carlson et al., 2025) dataset. We randomly selected 100,000 vectors to construct the index, with 100 queries and the top-k set to 100 for each query. We evaluated the performance of IVFPQ and HNSW by testing various parameter combinations. The specific parameters are as follows:

- **IVFPQ-1:** nlist=2048, nprobe=32, m=16
- **IVFPQ-2:** nlist=2048, nprobe=32, m=32
- **IVFPQ-3:** nlist=4096, nprobe=32, m=16
- **IVFPQ-4:** nlist=2048, nprobe=64, m=16
- **HNSW-1:** M=16, efConstruction=200, efSearch=2000
- **HNSW-2:** M=32, efConstruction=200, efSearch=1500
- **HNSW-3:** M=16, efConstruction=200, efSearch=1500

### EXPERIMENTAL RESULTS

The experimental results are shown in Table 5. The Flat Index has a shorter construction time but longer query time, though it supports lossless retrieval. IVFPQ, on the other hand, has a longer construction time but extremely short query times. HNSW, both in terms of construction and query

Table 5: Comparison of three Faiss indexes in terms of latency, memory, and accuracy.

| Method | Build Latency (s) | Search Latency (ms/query) | Memory Usage (MB) | Accuracy (%) |
|---|---|---|---|---|
| Flat | 0.03 | 0.577 | 49.12 | 100.00 |
| IVFPQ-1 | 7.27 | 0.018 | 57.12 | 81.93 |
| IVFPQ-2 | 8.25 | 0.025 | 82.49 | 83.89 |
| IVFPQ-3 | 14.19 | 0.016 | 67.75 | 81.92 |
| IVFPQ-4 | 7.49 | 0.020 | 54.00 | 82.63 |
| HNSW-1 | 0.39 | 0.068 | 49.05 | 81.65 |
| HNSW-2 | 0.43 | 0.038 | 73.21 | 82.69 |
| HNSW-3 | 0.81 | 0.038 | 48.53 | 81.57 |

times, exhibits relatively shorter durations. The memory cost of all three methods is comparable. In our open-source code, the construction method is set as a hyperparameter, allowing for flexibility in choosing the most suitable approach based on the specific use case.

# E EXTENDED EXPERIMENTS

## E.1 ADDITIONAL LATENCY ANALYSIS EXPERIMENTS

In the main paper, we mentioned that retrieval-based methods incur relatively high latency, but we did not provide detailed numbers. Therefore, we designed an experiment similar to Table 1 to analyze the latency of **PQCache**(Zhang et al., 2025), and the results are shown in Table 6.

Table 6: Latency comparison of **RACC** and **PQCache**.

| Method | TTFT (ms) | OGL with Different Generated Tokens | | |
|---|---|---|---|---|
| | | 512 (s) | 1024 (s) | 2048 (s) |
| Full Cache | 254 | 15.40 | 30.71 | 63.88 |
| SqueezeAttn | 515 | 46.83 | 96.09 | 186.75 |
| RACC | 259 | 15.96 | 32.40 | 65.66 |

The average generation speed of **PQCache** is approximately 10.967 tokens/s. Meanwhile, another retrieval-based method mentioned in the main paper, **RetrievalAttention**(Liu et al., 2024), is currently not open-sourced, so we cannot reproduce its implementation for experiments. However, its paper provides a clear number: around 0.188 s/token, i.e., 5.31 tokens/s. The standard inference speed is about 30 tokens/s, indicating that retrieval-based methods incur significantly higher latency compared to conventional compression methods.

## E.2 RESULTS AT DIFFERENT COMPRESSION RATIOS

In the main text, we compared the performance of **RACC** and baseline methods using a single compression ratio of 15%. Here, we additionally report the performance of each method at 20% and 30% compression ratios, tested using the Short Experiment of **LongGenBench**(Wu et al., 2025). The experimental results are shown in Table 7.

The experimental results demonstrate that RACC still achieves the best performance at other compression ratios.

Table 7: Comparison between RACC and Baselines at Other Compression Ratios

| Method | Compression Ratios=20% | | | Compression Ratios=30% | | |
|---|---|---|---|---|---|---|
| | Completion Rate | Single Accuracy | Range Accuracy | Completion Rate | Single Accuracy | Range Accuracy |
| Full Cache | 66.0 | 34.7 | 58.2 | 66.0 | 34.7 | 58.2 |
| SqueezeAttn | 62.4 | 30.1 | 52.5 | 60.5 | 31.8 | 53.3 |
| Scope | 54.1 | 31.3 | 52.6 | 53.7 | 32.7 | 54.9 |
| CakeKV | 61.8 | 27.6 | 45.3 | 63.5 | 28.9 | 49.3 |
| RACC | 64.9 | 32.1 | 54.2 | 64.7 | 33.3 | 55.1 |

### E.3 HYPERPARAMETER SENSITIVITY ANALYSIS

According to Eq. 4, $L_n$ is the compression window size, $\alpha$ is the window compression rate, $\beta$ is the retrieval budget, and $P_R$ is the overall compression rate.

These three hyperparameters ($L_n$, $\alpha$, $\beta$) can all affect the final generation accuracy by indirectly influencing the compression rate. To analyze the impact of different parameters on RACC's performance under the same compression rate, we conducted additional experiments. The experimental results are shown in Table 8.

Table 8: Sensitivity Analysis of Hyperparameters ($L_n$, $\alpha$, $\beta$) on RACC Performance

| $L_n$ | $\alpha$ | $\beta$ | $P_R$ | Completion Rate | Single Accuracy | Range Accuracy | Average Accuracy |
|---|---|---|---|---|---|---|---|
| 400 | 60% | 1% | 15% | 65.8 | 31.6 | 54.5 | 43.05 |
| 400 | 60% | 5% | 20% | 66.8 | 31.8 | 54.4 | 43.1 |
| 540 | 60% | 1% | 20% | 64.5 | 31.7 | 54.1 | 42.9 |
| 400 | 70% | 1% | 20% | 64.9 | 32.1 | 54.2 | 43.15 |
| 540 | 70% | 5% | 30% | 65.7 | 33.2 | 54.8 | 44.0 |
| 400 | 80% | 1% | 30% | 64.7 | 33.3 | 55.1 | 44.2 |
| 830 | 60% | 1% | 30% | 65.9 | 33.1 | 54.8 | 43.95 |
| Full Cache | – | – | 100% | 66.0 | 34.7 | 58.2 | 46.45 |

The experimental results show that a larger retrieval budget and a higher window compression rate both improve generation accuracy, at the cost of increased GPU and CPU memory consumption. Increasing the window size $L_n$ also leads to higher GPU memory usage, but the resulting accuracy gain is relatively limited.

### E.4 ABLATION STUDY

To investigate the contributions of the compression module and the retrieval module to the final accuracy, we design the following ablation study. Since the retrieval module cannot function independently without the compression module, we divide the method into two variants: **RACC (without Retrieval)** and **RACC (with Retrieval)**. By comparing their latency and accuracy, we can quantify the individual impact of each module and understand their overall influence on system performance.

We evaluate accuracy using the Short Experiment of **LongGenBench**, with the retrieval budget $\beta$ set to 1%. The results are summarized in Table 9.

The experimental results in this section demonstrate that both the compression module and the retrieval module play essential roles in enhancing the model's long-text generation capability. The retrieval module further improves accuracy on top of the compression module while introducing only minimal latency overhead, whereas the compression module provides stable baseline performance with reduced memory consumption. The combination of these two modules achieves a favorable balance among memory usage, latency, and accuracy, thereby validating the necessity and effectiveness of our overall design.

Table 9: Latency and Accuracy Ablation of RACC.

**(a) Latency Comparison**

| Method | TTFT (ms) | OGL with Different Generated Tokens | | |
|---|---|---|---|---|
| | | 512 tokens (s) | 1024 tokens (s) | 2048 tokens (s) |
| Full Cache | 254 | 15.40 | 30.71 | 63.88 |
| RACC (w/o retrieval) | 259 | 15.71 | 32.12 | 64.80 |
| RACC (with retrieval) | 259 | 15.96 | 32.40 | 65.66 |

**(b) Accuracy Comparison**

| Method | Completion Rate (%) | Single Accuracy (%) | Range Accuracy (%) |
|---|---|---|---|
| Full Cache | 66.0 | 34.7 | 58.2 |
| RACC (w/o retrieval) | 67.1 | 28.41 | 50.3 |
| RACC (with retrieval) | 65.8 | 31.6 | 54.5 |

### E.5 GENERALIZATION TO NEW MODEL ARCHITECTURES

To validate the adaptability of our method to larger models and updated model architectures, we applied it to the Qwen3 series(Yang et al., 2025). We conducted additional experiments on **Qwen3-14B**, using the Short Experiments from **LongGenBench** as the benchmark. Table 10 shows the comparison of **RACC** with a 15% compression rate against the full cache in terms of accuracy.

Table 10: RACC performance on **Qwen3-14B**

| Method | Completion Rate (%) | Single Accuracy (%) | Range Accuracy (%) |
|---|---|---|---|
| Full Cache | 51.4 | 48.9 | 75.1 |
| RACC | 48.8 | 43.39 | 71.0 |

The experimental results demonstrate that our method exhibits strong generalizability and adaptability. It is effective not only on the models evaluated in the main experiments but also on larger and more recent model architectures, highlighting its broad applicability.

## F THE USE OF LARGE LANGUAGE MODELS(LLMS)

This work utilized the assistance of LLMs solely for translation and language polishing. The ideas and the writing of the manuscript were developed independently, without the use of LLMs. We take full responsibility for all content presented in this article.

