# OpenReview forum: "RACC: Retrieval-Augmented KV Cache Compression in Long-Context Generation"
_ICLR.cc/2026/Conference — Submitted to ICLR 2026_

### Official Review · Reviewer_pN4F · 2025-10-28

**Soundness:** 3
**Presentation:** 3
**Contribution:** 2
**Rating:** 4
**Confidence:** 4

**Summary:**

The paper RACC: Retrieval-Augmented KV Cache Compression in Long-Context Generation proposes a hybrid framework to address the trade-off between accuracy and efficiency in long-context LLM inference. RACC maintains a compressed subset of important KV pairs on GPU while asynchronously retrieving potentially useful evicted tokens from CPU memory using maximum inner product search. Experimental results show that RACC achieves near-lossless accuracy with only 15% of the original KV cache, reducing latency to near full-cache levels and improving accuracy by 3–10% when combined with existing compression methods.

**Strengths:**

1. Clear motivation and solid problem framing: The paper addresses a critical and practical bottleneck in long-context LLM inference. The motivation is well grounded and directly linked to real-world deployment constraints.
2. Strong empirical performance: RACC consistently achieves near-lossless generation while using only ~15% of the original KV cache, with accuracy improvements of 3–10% when integrated with existing compression methods.
3. Theoretical grounding: The reduction of MIPS retrieval to nearest-neighbor search provides a clear mathematical foundation for the retrieval design, and the analysis of different retrieval indices adds depth

**Weaknesses:**

1. Limited novelty beyond integration: The core idea is conceptually straightforward and has been implicitly discussed in prior works (e.g., PQCache with approximate retrieval, SnapKV with selective KV maintenance)
2. Unclear retrieval efficiency vs. accuracy trade-off: The retrieval module relies on approximate MIPS, but the paper does not quantify how retrieval accuracy impacts generation quality.
3. Lack of principled design for hybrid coordination: The interaction between the compression module and retrieval module appears heuristic rather than theoretically justified or learned.

**Questions:**

1. How does RACC fundamentally differ from simply adding a retrieval buffer on top of an existing compression method?
2. Would the improvement still hold if retrieval results are imperfect or if the vector database grows large?
3. Can the authors clarify how their asynchronous retrieval differs from prefetch or speculative retrieval mechanisms proposed before?
4. Is there any formal analysis or ablation explaining why the specific design heuristics are optimal or robust?

---

> ### Author Response · Authors · 2025-11-21
> **Authors' response to Reviewer pN4F**
>
> Dear Reviewer,
>
> Thank you very much for your valuable comments on our paper. Your feedback has been tremendously helpful for our revisions. We address each of your concerns below:
>
> **1. Novelty of RACC**
>
> While RACC may appear similar to adding a retrieval buffer on top of existing compression methods, this is a combination we pioneered. Moreover, we improved the compression method and designed a rigorous asynchronous integration that incurs no additional time overhead. Furthermore, the term "retrieval buffer" is not entirely accurate—RACC leverages the redundant computational capacity of the CPU to correct the GPU's inference process, which is beyond what a simple "buffer" can achieve.
>
> **2. Details of Retrieval**
>
> The accuracy of approximate retrieval cannot be quantified on a case-by-case basis. Therefore, we conducted ablation experiments to evaluate the contributions of the compression and retrieval components. The results are as follows:
>
> | Method              | Completion Rate | Single Accuracy | Range Accuracy |
> |---------------------|-----------------|-----------------|----------------|
> | Full Cache          | 66              | 34.7            | 58.2           |
> | RACC (no retrieval) | 67.1            | 28.41           | 50.3           |
> | RACC (retrieval)    | 65.8            | 31.6            | 54.5           |
>
> This demonstrates that the retrieval mechanism is consistently effective.
>
> The retrieval module remains effective even with large vector databases. The LongBench dataset used in our paper includes prompts with lengths up to 1M tokens, which far exceeds the context length of most mainstream models. This proves that RACC remains effective under extreme conditions.
>
> **3. Distinction from Prior Work**
>
> Other retrieval methods, such as the prefetching strategy proposed by PQCache [1], use approximate attention to select approximate KV pairs, which are then used for approximate computation. This requires a number of retrieval operations equal to the number of layers for each generated token, resulting in unacceptable latency for practical deployment. In contrast, our approach performs retrieval in parallel with the standard inference pipeline to correct errors caused by compression, enabling online inference without introducing prohibitive time overhead.
>
> [1] Zhang, H., Ji, X., Chen, Y., Fu, F., Miao, X., Nie, X., Chen, W., & Cui, B. (2025). PQCache: Product Quantization-based KVCache for Long Context LLM Inference. arXiv preprint arXiv:2407.12820.
>
> **4. Why Our Framework is Robust**
>
> We approached this from a practical perspective. KVCache compression during inference has three objectives: low memory, high accuracy, and low latency. RACC was designed from the outset to achieve all three simultaneously. We referenced the "voting mechanism" from SnapKV and identified its limitations in scenarios such as short prompts with long generation. We innovatively proposed dynamic compression during the generation phase combined with vector retrieval to restore accuracy. Additionally, recognizing the non-negligible latency of vector retrieval, we innovatively proposed asynchronous querying to minimize inference latency. Therefore, our framework design is not heuristic but rather a feasible approximation to the optimal solution under well-defined inference objective constraints.
>
> ---
>
> We are continuing to refine our paper and look forward to your response and further discussion.

---

> ### Author Response · Authors · 2025-11-26
> **Authors' response to Reviewer pN4F sec2**
>
> Dear Reviewer,
>
> We previously provided several key technical details and additional experimental results, and we hope these revisions have addressed your concerns. If any questions remain, we would greatly appreciate hearing from you at your earliest convenience so that we may further refine the manuscript. As the discussion period is approaching its end, we look forward to your prompt response.

---

### Official Review · Reviewer_J8GA · 2025-10-30

**Soundness:** 3
**Presentation:** 3
**Contribution:** 2
**Rating:** 4
**Confidence:** 3

**Summary:**

This paper introduces RACC, a hybrid KV-cache management framework, which combines a compression-based method with a retrieval-based method. The core innovation lies in its asynchronous design, leveraging previously generated tokens as queries for retrieval to avoid stalling the GPU, and its scenario-aware compression strategies. The paper demonstrates that RACC can maintain near-lossless accuracy using only 15% of the original KV cache and can be orthogonally applied to other compression methods to boost their performance.

**Strengths:**

1. The core idea of augmenting a fast-but-lossy compression method with a slow-but-accurate retrieval method is effective and the empirical results support this, showing a superior accuracy-latency trade-off compared to compression-only baselines.
2. The decision to use the previously generated token as a query for retrieval, rather than the current one, is a novel insight. This asynchronous design is the key to hiding the retrieval latency and is a major differentiator from synchronous retrieval-based methods.
3. RACC's retrieval module can be plugged into existing compression methods like SnapKV and PyramidKV to improve their accuracy.
4. The paper provides a solid evaluation of latency, including Time-To-First-Token (TTFT) and Overall Generation Latency (OGL) across different output lengths (Table 1).

**Weaknesses:**

1. The chosen baselines (Scope, CakeKV, SqueezeAttn) are not the strongest or most relevant points of comparison. The field has recently seen several high-profile works that are directly comparable to RACC.
2. Section 4.2 discusses MIPS and its transformation to NNS, but it is never explicitly stated which specific retrieval algorithm is used in the main RACC experiments. The extensive evaluation of Flat, IVFPQ, and HNSW in Appendix D lacks full method description.
3. In Table 4, the metric is simply "Score." Without a definition (e.g., ROUGE, BLEU, a task-specific metric), these numbers are uninterpretable.
4. The paper does not detail the synchronization mechanism between the CPU and GPU. How many tokens ahead is the retrieval? What happens if the retrieval for token t isn't complete by the time token t+1 is being generated?
5. The paper does not include an ablation study to isolate the contribution of the retrieval component. How much does accuracy improve when adding retrieval to a base compression method? What is the latency overhead of the retrieval module in isolation?
6. The performance of RACC likely depends on the compression ratio α, the grouping frequency n, and the retrieval budget β. A sensitivity analysis on these key hyperparameters is missing.
7. The introduction and related work sections do a poor job of motivating why this specific hybrid approach is needed now. The critique of existing methods is surface-level.

**Questions:**

The same as the weakness.

**Details Of Ethics Concerns:**

No Ethics Concerns.

---

> ### Author Response · Authors · 2025-11-25
> **Authors' response to Reviewer  J8GA sec1**
>
> Dear Reviewer,
>
> Thank you very much for your valuable comments, which have been extremely helpful in revising our paper. We will address each of your concerns point by point:
>
> **1. Regarding Question 1: Adding more relevant baselines.** We have recently conducted an extensive review of related literature and found several works with high relevance to RACC. Among them, RocketKV [1] achieves KV cache compression through GPU-CPU collaboration. We designed experiments to compare latency and accuracy with this method, and the results are as follows:
>
> **Table 1: Comparison with RocketKV**
>
> | Method | 1token (ms) | 512tokens | 1024tokens | 2048tokens (s) |
> |--------|------------|-----------|------------|----------------|
> | Full Cache | 254 | 15.4 | 30.71 | 63.88 |
> | RocketKV | 447 | 25.96 | 52.6 | 105.33 |
> | RACC | 259 | 15.96 | 32.4 | 65.66 |
>
> | Method | Completion Rate | Single Accuracy | Range Accuracy |
> |--------|----------------|-----------------|----------------|
> | Full Cache | 66 | 34.7 | 58.2 |
> | RocketKV | 52.4 | 30.5 | 47.3 |
> | RACC | 65.8 | 31.6 | 54.5 |
>
> [1] Payman Behnam, Yaosheng Fu, Ritchie Zhao, Po-An Tsai, Zhiding Yu, and Alexey Tumanov. "RocketKV: Accelerating Long-Context LLM Inference via Two-Stage KV Cache Compression." arXiv preprint arXiv:2502.14051 (2025).
>
> The experimental results demonstrate that RACC still exhibits superiority compared to this more recent work.
>
> **2. Regarding Question 2: Detailed explanation of the index.** In our experiments, we use a flat index, implemented using the open-source Faiss library for retrieval. Detailed descriptions of various indexing methods can be found at https://faiss.com.cn/index.html.
>
> **3. Regarding Question 3: Confusion about the "score" concept in Table 4(paper).** Following the official LongBench evaluation methodology, we uniformly refer to the accuracy metric as "score," which is specifically calculated using ROUGE-L F1. We will add a detailed explanation of this in the paper.
>
> **4. Regarding Question 4: Confusion about the synchronization mechanism.** We set the window size to $L_n$. Within this window, the GPU is responsible for generating $L_n$ tokens sequentially, while the CPU performs incremental index construction and queries key vectors on the KV cache pruned from the previous window. At the end of the window, the GPU begins compression and concatenates the vectors retrieved by the CPU to the KV cache. We allocate sufficient time for retrieval (through a larger $L_n$) to ensure the smooth operation of the pipeline. In our experiments, we set $L_n$ to 400. For sensitivity analysis of $L_n$, please see Question 6.
>
> **5. Regarding Question 5: Ablation study.** We conducted ablation experiments on each component of RACC to analyze latency and accuracy. The experimental results are as follows:
>
> **Table 2: Ablation Study with Accuracy Verification Experiments**
>
> | Method | 1token (ms) | 512tokens | 1024tokens | 2048tokens (s) |
> |--------|------------|-----------|------------|----------------|
> | Full Cache | 254 | 15.4 | 30.71 | 63.88 |
> | RACC (no retrieval) | 259 | 15.71 | 32.12 | 64.8 |
> | RACC (retrieval) | 259 | 15.96 | 32.4 | 65.66 |
>
> | Method | Completion Rate | Single Accuracy | Range Accuracy |
> |--------|----------------|-----------------|----------------|
> | Full Cache | 66 | 34.7 | 58.2 |
> | RACC (no retrieval) | 67.1 | 28.41 | 50.3 |
> | RACC (retrieval) | 65.8 | 31.6 | 54.5 |
>
> **Due to the character limit, our response will continue in the next section.**

---

> > ### Author Response · Authors · 2025-11-25
> > **Authors' response to Reviewer  J8GA sec2**
> >
> > **6. Regarding Question 6:** In the original paper, we proposed the following formula to calculate the compression rate:
> >
> > $$P_R = \frac{\alpha^n - 1}{n(\alpha - 1)} + \beta$$
> >
> > where $n$ is the compression frequency
> > $$n = \left\lfloor \frac{\text{Generated tokens}}{L_n} \right\rfloor$$
> >
> > $L_n$ is the compression window size, $\alpha$ is the window compression rate, $\beta$ is the retrieval budget, and $P_R$ is the overall compression rate.
> >
> > All three hyperparameters ($L_n$, $\alpha$, $\beta$) can affect the final generation accuracy by indirectly influencing the compression rate. To analyze the impact of different parameters on RACC's performance under the same compression rate, we recently conducted additional experiments. Partial results are as follows:
> >
> > **Table 3: Sensitivity Analysis of Hyperparameters**
> >
> > | $L_n$ |  $\alpha$ | $\beta$ |$P_R$ | Completion Rate | Single Accuracy | Range Accuracy | Average Accuracy |
> > |-----|------------------------|------------------|-----|----------------|-----------------|----------------|-------------------|
> > | 400 | 60% | 1% | 15% | 65.8 | 31.6 | 54.5 | 43.05 |
> > | 400 | 60% | 5% | 20% | 66.8 | 31.8 | 54.4 | 43.1 |
> > | 540 | 60% | 1% | 20% | 64.5 | 31.7 | 54.1 | 42.9 |
> > | 400 | 70% | 1% | 20% | 64.9 | 32.1 | 54.2 | 43.15 |
> > | 540 | 70% | 5% | 30% | 65.7 | 33.2 | 54.8 | 44 |
> > | 400 | 80% | 1% | 30% | 64.7 | 33.3 | 55.1 | 44.2 |
> > | 830 | 60% | 1% | 30% | 65.9 | 33.1 | 54.8 | 43.95 |
> > | - | Full Cache | - | 100% | 66 | 34.7 | 58.2 | 46.45 |
> >
> > The experimental results indicate that a larger retrieval budget and a larger window compression rate can both improve generation accuracy, at the cost of requiring more GPU and CPU memory. Increasing the window size $L_n$ also increases GPU memory usage, but the accuracy improvement is relatively small.
> >
> > **7. Regarding Question 7: Wording issues.** Thank you for your suggestion. We will revise the related work section in the final version of the paper.
> >
> > We are currently optimizing the paper, and all the above experimental results will be incorporated. We look forward to your response and further discussion with you.

---

> > > ### Author Response · Authors · 2025-11-28
> > >
> > > Dear Reviewer,
> > >
> > > Thank you again for your careful consideration of our work. In our previous revision, we added several important technical clarifications and supplementary experimental results in response to your comments. We hope that these updates have addressed the concerns you raised.
> > >
> > > If there are any remaining questions or points that require further revision, we would sincerely appreciate your feedback at your earliest convenience, as we are fully committed to improving the manuscript. Since the discussion period is nearing its conclusion, we would be grateful for your timely response.
> > >
> > > Thank you very much for your time and effort.

---

### Official Review · Reviewer_pvML · 2025-10-30

**Soundness:** 3
**Presentation:** 3
**Contribution:** 3
**Rating:** 4
**Confidence:** 5

**Summary:**

This paper introduces RACC, a novel hybrid framework to address the significant KV cache bottleneck in long-context LLM inference. The method combines GPU-side KV cache compression with an efficient, asynchronous retrieval mechanism. RACC evicts KVs to a CPU-managed vector index and retrieves them asynchronously. This retrieval is guided by queries from previously generated tokens, leveraging the observation that adjacent tokens share similar attention patterns. The goal is to achieve the near-lossless accuracy of full-context models while maintaining the low-latency performance of compression-only methods.

**Strengths:**

1) The combination of compression and retrieval approaches is conceptually sound, addressing the limitations of each individual approach (accuracy degradation in compression methods and latency increase in retrieval methods).

2) The asynchronous CPU-side retrieval mechanism is clever, leveraging the similarity between adjacent tokens' attention distributions to avoid blocking GPU inference, thus maintaining low latency.

3) RACC demonstrates impressive performance, maintaining accuracy close to full-cache methods while using only 15% of the original KV cache.

4) The framework shows good compatibility with existing compression methods like SnapKV and PyramidKV, enhancing their performance when integrated with RACC.

**Weaknesses:**

1. Inadequate Baseline Comparisons

The paper fails to include critical baselines necessary to substantiate its claims. It positions RACC as a superior hybrid solution over pure compression and pure retrieval methods, yet experimental comparisons (Tables 1–3) only cover compression-based methods (e.g., Scope, CakeKV, SqueezeAttn) and omit direct comparisons with retrieval-based methods (e.g., PQCache) discussed in the related work—leaving the claim that RACC avoids "huge inference latency" of retrieval methods unsubstantiated.

Additionally, RACC is not thoroughly compared against other hybrid memory management approaches for LLMs, such as recent works exploring CPU-GPU collaborative inference strategies, which would serve as more appropriate baselines for its hybrid design.

2. Limited Experimental Design and Generalization

Narrow model scope: Experiments are restricted to only two 8B-scale models (LLAMA-3.1-8B, Mistral-7B-v0.2) with no validation on larger models (e.g., 70B-scale or above) where memory constraints are more severe—undermining claims of generality.

Unjustified compression ratio: A single compression ratio (15%) is used across all experiments, with no justification for this specific value or exploration of performance trends under different compression ratios.

Missing ablation for key components: The paper introduces three adaptive compression strategies (LISO, SILO, LILO) for different tasks but lacks a clear ablation study. It neither specifies which strategy was applied to each benchmark (LongGenBench, LongProc) nor provides comparative analysis to demonstrate why this adaptive approach outperforms a single naive strategy (e.g., using SILO for all tasks)—failing to validate this design choice.

3. Insufficient Methodological Novelty, Details, and Clarity

Limited novelty: RACC essentially combines two existing approaches (compression and retrieval). While its asynchronous retrieval mechanism is clever, it builds directly on established vector search techniques and well-documented attention distribution properties from prior work, offering limited novel contributions.

Missing critical details: Key technical details are insufficiently described, including the CPU-GPU communication mechanism, vector index construction process, and overhead of building/maintaining the vector index (a gap尤为 notable given the paper’s focus on inference performance).

Poor component placement: The "voting mechanism"—a core part of the compression module—is relegated to Appendix B. Given its importance to RACC’s operation, a complete description should be integrated into the main body (Section 4.3) to improve clarity.

4. Incomplete Latency Analysis

The paper’s latency analysis is inadequate, as it does not fully account for potential variability in CPU-GPU communication overhead under different workloads or hardware configurations. While it claims minimal latency impact, it fails to investigate worst-case scenarios or performance degradation under resource contention (e.g., when batch size > 1).

5. Unaddressed Scalability and Applicability Limitations

RACC’s scalability to broader scenarios is not addressed: the paper does not adequately explain how it would perform with longer contexts (>64K tokens) or scale to much larger models—key use cases where memory-efficient solutions are most critical.

**Questions:**

see weakness part

---

> ### Author Response · Authors · 2025-11-21
> **Authors' response to Reviewer pvML  sec1**
>
> Dear Reviewer,
>
> Thank you very much for your comments, which have been tremendously helpful. We address each of your concerns below:
>
> **1. Insufficient Baseline Comparisons**
>
> We mentioned in the paper that retrieval-based methods introduce significant latency but did not provide specific experimental results. We have now supplemented this with the following experiments:
>
> | Method   | 1 token (ms) | 512 tokens | 1024 tokens | 2048 tokens |
> |----------|--------------|------------|-------------|-------------|
> | Full Cache | 254        | 15.4       | 30.71       | 63.88       |
> | PQCache  | 515          | 46.83      | 96.09       | 186.75      |
> | RACC     | 259          | 15.96      | 32.4        | 65.66       |
>
> Unfortunately, another retrieval-based method we mentioned, RetrievalAttention [1], is not open-sourced, so we cannot provide additional data. However, according to the latency data reported in their paper (0.188s/token, i.e., 5.31 tokens/s), this is significantly slower than normal inference speed (30 tokens/s).
>
> We have recently conducted an extensive literature review and identified several works highly relevant to RACC. Among them, RocketKV [2] achieves KVCache compression through GPU-CPU collaboration. We designed experiments to compare against it, with partial results shown below:
>
> | Method     | 1 token (ms) | 512 tokens | 1024 tokens | 2048 tokens (s) |
> |------------|--------------|------------|-------------|-----------------|
> | Full Cache | 254          | 15.4       | 30.71       | 63.88           |
> | RocketKV   | 447          | 25.96      | 52.6        | 105.33          |
> | RACC       | 259          | 15.96      | 32.4        | 65.66           |
>
> | Method     | Completion Rate | Single Accuracy | Range Accuracy |
> |------------|-----------------|-----------------|----------------|
> | Full Cache | 66              | 34.7            | 58.2           |
> | RocketKV   | 52.4            | 30.5            | 47.3           |
> | RACC       | 65.8            | 31.6            | 54.5           |
>
> [1] Liu, D., Chen, M., Lu, B., Jiang, H., Han, Z., Zhang, Q., Chen, Q., Zhang, C., Ding, B., Zhang, K., Chen, C., Yang, F., Yang, Y., & Qiu, L. (2024). RetrievalAttention: Accelerating Long-Context LLM Inference via Vector Retrieval. arXiv preprint arXiv:2409.10516.
>
> [2] Behnam, P., Fu, Y., Zhao, R., Tsai, P.-A., Yu, Z., & Tumanov, A. (2025). RocketKV: Accelerating Long-Context LLM Inference via Two-Stage KV Cache Compression. arXiv preprint arXiv:2502.14051.
>
> **2. Limited Experimental Design and Generalization**
>
> In our paper, we conducted experiments on LLaMA 8B and Mistral 7B because all selected baselines also performed experiments on these models, which we believe is sufficient to demonstrate RACC's superiority. However, some new model architectures have recently emerged, such as Qwen3, for which these baselines have not yet provided optimization support. We have recently added compatibility for the Qwen3 series and selected the larger Qwen3 14B for experiments. The observed results are as follows:
>
> | Method     | Completion Rate | Single Accuracy | Range Accuracy |
> |------------|-----------------|-----------------|----------------|
> | Full Cache | 51.4            | 48.9            | 75.1           |
> | RACC       | 48.8            | 43.39           | 71             |
>
> The above results demonstrate that our method remains effective on larger models and other architectures.
>
> Regarding the compression ratio, we have recently supplemented experiments at 20% and 30% compression rates. Partial results are shown below:
>
> | Method     | Completion Rate | Single Accuracy | Range Accuracy |
> |------------|-----------------|-----------------|----------------|
> | Full Cache | 66              | 34.7            | 58.2           |
> | Scope(20%) | 54.1            | 31.3            | 52.6           |
> | Scope(30%) | 53.7            | 32.7            | 54.9           |
> | RACC(20%)  | 64.9            | 32.1            | 54.2           |
> | RACC(30%)  | 64.7            | 33.3            | 55.1           |
>
> **3. Implementation Details and Writing Issues**
>
> Thank you very much for your guidance on paper writing. We will incorporate your suggestions to revise our paper. Regarding implementation details (GPU-CPU communication mechanism and communication latency ablation experiments): All our work is implemented using the PyTorch framework, which enables straightforward GPU-CPU communication. For specific technical details, please refer to https://docs.pytorch.org/tutorials/intermediate/pinmem_nonblock.html. The ablation experiments will be addressed together with Question 4.
>
> **Due to the character limit, our response will continue in the next section.**

---

> > ### Author Response · Authors · 2025-11-21
> > **Authors' response to Reviewer pvML sec2**
> >
> > **4. Incomplete Latency Analysis**
> >
> > Both RACC and all baseline methods use the Transformers framework for inference, which does not yet support batch inference. GPU-CPU communication implemented via PyTorch demonstrates strong stability. Additionally, we have conducted ablation experiments to analyze the latency of the retrieval component, with results shown below:
> >
> > | Method              | 1 token (ms) | 512 tokens | 1024 tokens | 2048 tokens（s） |
> > |---------------------|--------------|------------|-------------|-------------|
> > | Full Cache          | 254          | 15.4       | 30.71       | 63.88       |
> > | RACC (no retrieval) | 259          | 15.71      | 32.12       | 64.8        |
> > | RACC (retrieval)    | 259          | 15.96      | 32.4        | 65.66       |
> >
> > Compared to inference without retrieval compression, the additional latency from the retrieval component consists of extra tensor processing and GPU-CPU communication. The experiments demonstrate that the latency introduced by GPU-CPU communication is negligible in the overall inference pipeline.
> >
> > **5. Unaddressed Scalability and Applicability Limitations**
> >
> > In Section 5.4, we evaluated our approach using LongBench as the benchmark dataset, which includes prompts with lengths up to 1M tokens. Our experiments demonstrate that RACC operates smoothly and delivers optimizations even under such demanding conditions.
> >
> > ---
> >
> > We are continuing to refine our paper, and complete versions of all the above experiments will be incorporated. We look forward to your response and further discussion.

---

> ### Author Response · Authors · 2025-11-26
> **Authors' response to Reviewer pvML sec3**
>
> Dear Reviewer,
>
> Thank you for your positive assessment of our work, particularly your “good” ratings on the soundness, presentation, and contribution of the paper. Following your suggestions, we have conducted additional experiments and provided further technical details in the discussion section. We hope these revisions address your concerns. If any issues remain unresolved, please feel free to share additional feedback—we are committed to further improving the manuscript. We look forward to your response.

---

### Official Review · Reviewer_GWHc · 2025-10-31

**Soundness:** 4
**Presentation:** 3
**Contribution:** 3
**Rating:** 6
**Confidence:** 3

**Summary:**

This paper presents RACC, a hybrid KV cache framework that addresses the accuracy-latency trade-off in long-context inference by combining GPU-side compression with CPU-side retrieval. Its core innovation is an asynchronous retrieval mechanism that uses previously generated tokens as queries, leveraging their attention similarity to current tokens to run retrieval in parallel with GPU decoding and avoid I/O bottlenecks. RACC achieves near-lossless accuracy using only 15% of the original KV cache, demonstrating superior performance over existing compression-only and retrieval-only methods.

**Strengths:**

1. The core innovation lies in using query vectors from previously generated tokens to conduct asynchronous CPU-side retrieval. This design adeptly circumvents the severe latency bottleneck caused by synchronous I/O, which plagues traditional retrieval-based methods.

2. RACC successfully breaks the longstanding compromise between inference efficiency and generation accuracy. It retains the high efficiency and low memory footprint of compression on the GPU while using asynchronous retrieval to "recall" crucial evicted information, thus maintaining high fidelity.

3. The experimental results are compelling. RACC delivers near-lossless performance with only 15% of the KV cache. Under the same compression ratio, it significantly outperforms SOTA baselines like SqueezeAttn, Scope, and CakeKV in generation accuracy on benchmarks like LongGenBench and LongProc.

**Weaknesses:**

1. High Dependence on the Core Observation: The method's efficacy hinges entirely on the observation that "adjacent tokens share similar attention patterns". For tasks requiring rapid, sharp context shifts or the precise retrieval of a single, non-redundant fact from the distant past, using a stale query ($q_{t-n}$) may fail to retrieve the exact KV pairs needed by the current token ($q_t$)

2. Increased System Complexity: The framework introduces non-trivial system complexity. It requires maintaining a persistent vector database on the CPU and managing sophisticated, error-free asynchronous communication and data transfer between the CPU and GPU.

3. Unaddressed Dynamic Indexing Costs: For SILO and LILO scenarios, the paper states the CPU-side vector index is "incrementally updated". However, the paper does not analyze the computational overhead of these dynamic index updates or the potential for the MIPS search itself to become a new bottleneck as the index grows to manage millions of evicted tokens.

**Questions:**

1. Query Lag Sensitivity: What is the exact lag used in the experiments between the query token $q$ and the token $t$ currently being generated (e.g., is it $q_{t-1}$ or $q_{t-n}$)? How sensitive is RACC's performance (both accuracy and latency) to this "query staleness" or lag?

2. Dynamic Index Bottleneck: In LILO scenarios , the CPU index must grow continuously as tokens are evicted. At what point (e.g., after how many generated tokens) does the CPU-side MIPS search and/or incremental index updating become the new performance bottleneck, potentially negating the latency gains from asynchronous execution?

3. Grouping Compression Parameters: The paper introduces a "grouping compression strategy" (compressing every n new tokens) to reduce computational overhead. What value of n was used in the experiments? How was the trade-off between reducing compression frequency (a larger n) and ensuring the GPU cache remains sufficiently relevant (a smaller n) determined?

---

> ### Author Response · Authors · 2025-11-21
> **Authors' response to Reviewer GWHc**
>
> Dear Reviewer,
>
> Thank you for your valuable feedback and recognition of our work. We address each of your concerns below:
>
> **1. Query Lag Sensitivity**
>
> We set the distance between the query vector and the currently generated token (hereafter referred to as "distance") to 50. This value is determined based on two key considerations:
>
> First, as stated in our paper:
>
> > "We observe that adjacent windows share similar attention allocation patterns, whereas this similarity decreases as the window distance increases."
>
> Tokens in closer proximity exhibit more similar attention allocation patterns, which constrains the distance from being too large.
>
> Second, due to the parallelization of querying and inference, an excessively short distance would result in the inference window completing before the query finishes, thereby disrupting our framework design. Therefore, the distance cannot be too small either.
>
> The distance can certainly be set to larger values, as it is implemented as a hyperparameter in our system. We conducted sensitivity experiments, and the results are presented below:
>
> | Distance | Completion Rate | Single Accuracy | Range Accuracy |
> |----------|-----------------|-----------------|----------------|
> | 50       | 65.8            | 31.6            | 54.5           |
> | 100      | 64.2            | 30.8            | 53.7           |
> | 150      | 64.8            | 30.5            | 52.9           |
>
> **2. Dynamic Index Bottleneck**
>
> An excessively large vector database may introduce increased query latency and incremental construction overhead, potentially creating new bottlenecks. In Section 5.4, we evaluated our approach using LongBench as the benchmark dataset, which includes prompts with lengths up to 1M tokens. Our experiments demonstrate that RACC operates smoothly and delivers optimizations even under such demanding conditions. Since a context length of 1M tokens far exceeds the context window of most mainstream models, this does not constitute an additional bottleneck.
>
> **3. Grouped Compression Parameters**
>
>   We have revised some of the notations in the paper. We set the window size of the "group query strategy" (In the original paper, this quantity was denoted as $n$; it is now denoted as $L_n$.) to 400. The model's generation speed is approximately 30 tokens/s. With $L_n=400$, we can ensure sufficient time for the database to complete incremental construction and querying. A larger $L_n$ represents low-frequency compression, retaining more KV cache; a smaller $L_n$ represents high-frequency compression, retaining less KV cache. The setting of window $L_n$ depends on the KV cache budget.
>
> We have supplemented sensitivity analysis experiments for various hyperparameters appearing in the paper. Before that, I will reintroduce the role of each hyperparameter. According to the formula proposed in our paper:
>
> $$P_R = \frac{\alpha^n - 1}{n(\alpha - 1)} + \beta$$
>
> where $n$ is the compression frequency
> $$n = \left\lfloor \frac{\text{Generated tokens}}{L_n} \right\rfloor$$
>
> $L_n$ is the compression window size, $\alpha$ is the window compression rate, $\beta$ is the retrieval budget, and $P_R$ is the overall compression rate.
>
> All three hyperparameters ($L_n$, $\alpha$, $\beta$) can affect the final generation accuracy by indirectly influencing the compression rate. To analyze the impact of different parameters on RACC's performance under the same compression rate, we recently conducted additional experiments. Partial results are as follows:
>
> **Table 3: Sensitivity Analysis of Hyperparameters**
>
> | $L_n$ |  $\alpha$ | $\beta$ |$P_R$ | Completion Rate | Single Accuracy | Range Accuracy | Average Accuracy |
> |-----|------------------------|------------------|-----|----------------|-----------------|----------------|-------------------|
> | 400 | 60% | 1% | 15% | 65.8 | 31.6 | 54.5 | 43.05 |
> | 400 | 60% | 5% | 20% | 66.8 | 31.8 | 54.4 | 43.1 |
> | 540 | 60% | 1% | 20% | 64.5 | 31.7 | 54.1 | 42.9 |
> | 400 | 70% | 1% | 20% | 64.9 | 32.1 | 54.2 | 43.15 |
> | 540 | 70% | 5% | 30% | 65.7 | 33.2 | 54.8 | 44 |
> | 400 | 80% | 1% | 30% | 64.7 | 33.3 | 55.1 | 44.2 |
> | 830 | 60% | 1% | 30% | 65.9 | 33.1 | 54.8 | 43.95 |
> | - | Full Cache | - | 100% | 66 | 34.7 | 58.2 | 46.45 |
>
> The experimental results indicate that a larger retrieval budget and a larger window compression rate can both improve generation accuracy, at the cost of requiring more GPU and CPU memory. Increasing the window size $L_n$ also increases GPU memory usage, but the accuracy improvement is relatively small.
>
> ---
> We are currently optimizing the paper and look forward to your response and further discussion with you.

---

> ### Author Response · Authors · 2025-11-26
> **Authors' response to Reviewer GWHc sec2**
>
> Dear Reviewer,
>
> Thank you very much for your recognition of our work. We have provided detailed responses to all the questions you raised. If any of your concerns remain unaddressed, please feel free to let us know. We will further refine the manuscript accordingly. We look forward to hearing from you soon.

---

### Author Response · Authors · 2025-12-01
**Rebuttal Summary by Authors**

**Dear Area Chair,**

We hope this message finds you well, and we sincerely appreciate your time and effort in handling our submission.

As the discussion period is drawing to a close, and given that all exchanges during the rebuttal phase are publicly visible, we would like to provide some additional context to ensure you have as much relevant information as possible when making the final decision.

1. **Regarding our main contribution:**
   Existing KV-cache compression methods typically operate at the granularity of individual tokens and permanently discard those deemed “unimportant.” However, our experiments show that token importance is dynamic: tokens that appear unimportant early on may become essential at later stages of generation. Discarding them too early introduces an accumulating error that can cause long-text generation to deviate significantly. Our proposed RACC framework is the first to systematically address this issue, achieving both lower generation latency and higher accuracy compared with conventional methods.

2. **Regarding the discussion timeline:**
   We submitted our detailed responses to all reviewer comments shortly before the recent “incident.” As a result, we have not yet received follow-up replies from the reviewers. We believe this is mainly due to the limited amount of time rather than negative sentiment toward our work.

3. **Regarding reviewer feedback:**
   Reviewers GWHc, pvML, and J8GA provided high ratings in terms of Soundness, Presentation, and Contribution. In their Strengths sections, they repeatedly praised the design, integration, and theoretical rigor of our method. Their primary suggestions were related to additional experiments, all of which we have incorporated in the revised version.
   Reviewer pN4F’s comments mainly concerned the differences between our approach and some existing works. While we have discussed these distinctions thoroughly across multiple sections of the paper, we are concerned that certain details may not have been fully captured during their reading or that some technical background may not have been entirely clear, potentially leading to a degree of misunderstanding. We deeply respect every reviewer’s expertise and hope to clarify our contributions as clearly as possible within the limited discussion window.

Given the above, we wish to share this information candidly so that you may have a more comprehensive picture when making your final assessment. We truly appreciate your time, consideration, and assistance.

---

### Meta-Review · Area_Chair_M55B · 2026-01-07

**Summary:**

The paper tries to solve an important problem about KV cache compression. The rebuttal has introduced additional helpful experimental results; however, the main concern about the lack of novelty is not fully addressed. Thus, I have decided to reject the paper.

**Reviewer Concerns:**

The concerns about additional experimental results could be addressed, while those about the lack of novelty are not. To be more specific, the method essentially combines two existing approaches (compression and retrieval), and the author clearly explains the details of its methodology; yet this key issue still persists.

**Reviewer Scores:**

I tend to believe the reviewers would keep their original score.

---

### Decision · Program_Chairs · 2026-01-26

Reject